# Enhancing Protoplast Isolation and Early Cell Division from *Cannabis sativa* Callus Cultures via Phenylpropanoid Inhibition

**DOI:** 10.3390/plants13010130

**Published:** 2024-01-02

**Authors:** Adrian S. Monthony, Andrew Maxwell P. Jones

**Affiliations:** 1Département de Phytologie, Université Laval, Québec, QC G1V 0A6, Canada; 2Institut de Biologie Intégrative et des Systèmes (IBIS), Université Laval, Québec, QC G1V 0A6, Canada; 3Centre de Recherche et D’innovation sur les Végétaux (CRIV), Université Laval, Québec, QC G1V 0A6, Canada; 4Institut Intelligence et Données (IID), Université Laval, Québec, QC G1V 0A6, Canada; 5Department of Plant Agriculture, University of Guelph, Guelph, ON N1G 2W1, Canada

**Keywords:** protoplast, *Cannabis sativa*, regeneration, AIP, phenylpropanoid, tissue culture

## Abstract

De novo regeneration of *Cannabis sativa* L. (cannabis) using tissue culture techniques remains unreliable and infrequent. Conventional methods for the regeneration and transformation of cannabis have not achieved the reliability and replicability that need to be integrated into research and breeding programs. Protoplast systems are effective for gene expression studies and transformation and genome-editing technologies and open the possibility of somatic hybridization to create interspecific hybrids. To date, leaf-derived protoplasts have been isolated for transient gene expression studies, but protoplast-to-plant regeneration has not been reported. The present study aims to evaluate the efficacy of using a callus culture system as an abundant tissue source for protoplast isolation and lays the groundwork for a protoplast-to-plant regeneration system. Using hypocotyl-derived callus cultures, which are known to have relatively greater regenerative potential, the efficacy of protoplast isolation and initial cell division were assessed. In this study, the effect of 2-aminoindane-2-phosphonic acid (AIP), a competitive inhibitor of phenylalanine ammonia lyase (PAL), in callus culture media and the effect of subculture frequency on protoplast yield were assessed. This study found that inclusion of AIP at 1 mM resulted in a 334% increase in protoplast yield compared with AIP-free medium, representing the first known use of AIP in cannabis tissue culture. Inclusion of AIP led to a 28% decrease in total soluble phenolics and 52% decrease in tissue browning compared with the control medium. Lastly, a two-phase culture system for protoplast regeneration was tested. At a concentration of 2.0 × 10^5^ protoplasts per mL, cell wall reconstitution and cell division were observed, providing one of the first know reports of cell division from cannabis protoplasts and setting the stage for the future development of a protoplast-to-plant regeneration system.

## 1. Introduction

The global trend towards *Cannabis sativa* L. (cannabis) legalization has revealed significant challenges in its cultivation and the application of biotechnologies to the species. Plant tissue culture, offering advantages such as germplasm preservation and disease-free material propagation, holds promise for the industry and researchers [1]. However, success relies on developing robust methods that work across diverse genotypes [1,2]. Despite reports of high micropropagation and regeneration rates [2,3], attempts to replicate these reported methods have yielded inconsistent results, leading some researchers to label cannabis as recalcitrant [4,5,6]. 

Cannabis’ recalcitrance to regeneration has resulted in underdeveloped transformation systems for gene function studies or genetic improvement in the species. An alternative approach to studying gene function and expression dynamics is the use of plant protoplasts, where plant cells are stripped of their cell walls. The removal of the cell wall facilitates the introduction of foreign genetic material, thereby providing an optimal platform for transient gene expression studies or stable cell transformation/editing. Transformed/edited protoplasts can be utilized for whole-plant regeneration, and due to their single-cell origin, the regenerated plants are less likely to be chimeric than plants regenerated using other approaches [7]. Another valuable application of plant protoplasts is in somatic hybridization, where protoplasts from two individuals can be fused, and the resulting fusion product can be regenerated into a hybrid. Protoplast fusion offers an intriguing potential in cannabis: generating hybrids with enhanced disease resistance and creating new plant varieties for commercial or medicinal use. Such approaches have been employed in the citrus industry, where somatic hybridization by protoplast fusion has proved crucial for overcoming challenges posed by high heterozygosity, enabling the development of hybrids with desirable traits and even the fusion of sterile clones [8]. However, realizing these applications in cannabis requires a dependable protoplast-to-plant regeneration system, which has yet to be reported.

The practice of protoplast isolation and culture in cannabis remains in its initial stages. The first report of cannabis protoplast isolation came from Morimoto et al. [9]. However, these protoplasts were not cultured or regenerated. For over a decade, this report remained the sole mention of protoplast isolation in cannabis, but recently, several reports have emerged [7,10,11,12]. Beard et al. [7] demonstrated transient gene expression in cannabis leaf protoplasts using a high-cannabidiol (CBD) cultivar “Cherry × Otto II: Sweetened”, yet no efforts at culturing and regenerating protoplasts were reported in their study. Subsequent reports have introduced improved methods for protoplast isolation and transient gene expression from leaves [7,11,12], but these advances have focused solely on isolation and transient gene expression, omitting considerations for regeneration.

The choice of tissues for regeneration is crucial, with more responsive tissues like hypocotyls offering a potentially more viable avenue for achieving regeneration as compared with leaf tissues used in previous methods [13,14]. Protoplast isolation depends on several factors, with cell wall digestibility playing a critical role. Plant cell walls are predominantly composed of cellulose, but derive significant resilience from the presence of lignin [15,16,17] and other compounds found in the cell walls, such as ferulic and chlorogenic acid, which have been identified as digestion-interfering agents in protoplast isolation [18]. Strategies to weaken cell walls involve reducing lignin precursors produced by phenylalanine-ammonia lyase (PAL). Competitive enzyme inhibitors of PAL, such as 2-aminoindane-2-phosphonic acid (AIP), improve protoplast isolation efficiency by reducing monolignol subunits and improving cell wall digestibility [15,16,17]. Notably, AIP’s application improved protoplast yield and the subsequent regeneration of morphologically normal explants in American elm callus [19,20,21], when prior studies involving American elm protoplasts had failed to generate plants [22,23].

The focus of this study was to evaluate the effects of AIP on callus growth, protoplast isolation, and early cell division in *C. sativa* as a first step toward protoplast-to-plant regeneration in the species. The results demonstrate the benefits of AIP in improving protoplast isolation from cannabis callus and provide the first report of cell division from cannabis-derived protoplasts. Further work is needed to overcome the recalcitrant nature of cannabis, and this communication provides a valuable starting point for the development of protoplast-to-plant regeneration methods in the species. 

## 2. Results

### 2.1. Callus Culture Initiation, Digestibility, and Protoplast Yield

Callus cultures were successfully established on callus induction medium (LT-C; see Section 4 for composition) and callus induction medium + 1 mM AIP (LT-AIP; see Section 4 for composition) media within 2 months [6]. Routine isolations were performed with a weekly or biweekly frequency beginning after week 10, as described in Section 4. Callus growth remained robust throughout the experiment, with LT-C calli exhibiting a browner and firmer appearance compared with LT-AIP calli (Figure 1A). The analysis of variance (ANOVA) revealed that media significantly influenced protoplast yield (*p* < 0.03; Appendix A), with a 334% increase in protoplast yield for LT-AIP media compared with LT-C treatments being reported (5.39 × 10^5^ protoplasts/g vs. 1.24 × 10^5^ protoplasts/g; Figure 2). Microscopic examination of LT-C callus revealed poor digestion with no visible protoplasts amongst the large undigested cell clusters (Figure 1B). LT-AIP-derived callus digests revealed fewer large clusters of undigested cells and numerous single undigested cells alongside protoplasts. Neither the interaction between media and the subculture frequency (*p* = 0.1369) nor the subculture frequency (*p* = 0.9852) alone was found to have a significant effect on the yield (Appendix A).

### 2.2. Total Soluble Phenolics and Tissue Browning

The influence of media and subculture frequency on phenolic accumulation was assessed through measurements of total soluble phenolics and tissue browning. Total soluble phenolics, quantified as gallic acid equivalents (GAE) using the Folin–Ciocalteu (F-C) assay, were impacted by media type (*p* = 0.0053; Appendix A). The callus grown on LT-AIP medium exhibited an approximately 28% lower total soluble phenolic content compared with the LT-C callus (Figure 2B). Neither subculture frequency nor the interaction between media and subculture frequency affected total soluble phenolics (*p* > 0.05; Appendix A). Similarly, the 340 nm (OD340) assay indicated that AIP inclusion in the medium reduced browning (*p* < 0.001), with a 52% reduction in browning associated with phenolics observed in the LT-AIP treatment versus the LT-C-sourced callus (Figure 2B). Browning attributed to soluble phenolics was not influenced by subculture frequency or the interaction between media and subculture frequency (*p* > 0.05; Appendix A). Additionally, a Kendall tau-b ranked correlation (Appendix A) revealed a weak negative correlation between protoplast yield and total soluble phenolics determined by the F-C assay. Likewise, there was a weak negative correlation between yield and tissue browning. Both the F-C assay and tissue browning assay results demonstrated a moderate positive correlation with each other.

### 2.3. Protoplast Viability and Culture

Protoplasts for culture were obtained from callus digests with a 100 mg callus to 5 mL enzyme ratio, as described in Section 4. To ensure sufficient protoplast volumes, these digests were scaled up to 600 mg of fresh weight and 30 mL of enzyme. The protoplast yield from the LT-AIP-derived callus in the large digests surpassed that of the LT-C treatment by 206% (Table 1). Protoplast viability exceeded 90% in both treatments and showed no significant difference (Figure 1C and Table 1).

Due to the lower yield observed in the LT-C treatment, protoplasts were exclusively cultured from the LT-AIP treatment. Culturing experiments evaluated densities of 0.5 × 10^5^, 1.0 × 10^5^, and 2.0 × 10^5^ protoplasts/mL. Low-melting-point agarose beads were cultured in liquid KM5/5 medium (Kao and Michayluk; see Section 4) [24] with and without 10 µM AIP. Protoplasts cultured at a density of 2.0 × 10^5^ exhibited early signs of cell division, which were not observed at lower densities. Within 6 days of culture, indications of bulging cell membranes and the formation of peanut- and pear-shaped cells became evident, along with initial dividing cells (Figure 3A–D). After approximately three weeks of culture, small cell clusters (microcalli) had developed (Figure 3E,F). However, around the three-week mark, protoplast growth ceased, and cell viability declined. Protoplasts cultured in KM5/5 with 10 µM AIP at a density of 2.0 × 10^5^ appeared to initiate cell division sooner than those in media without AIP. Cultures grown at 0.5 × 10^5^ showed no response, while cultures at 1.0 × 10^5^ exhibited limited cell division that was not sustained beyond one week.

## 3. Discussion

This study marks a significant first step in establishing a protoplast-to-plant regeneration system and reports the first-known instance of early cell division in cannabis-derived protoplasts. Protoplast isolation in cannabis has received limited attention [7,9,10,11,14], with a predominant focus on leaf tissues for transient gene expression studies [7,11,14]. While leaf tissues are a convenient protoplast source [25,26], they may not consistently exhibit the highest regenerative capacity, as has been observed in American elm [18,20,22,23]. In contrast, protoplasts derived from elm callus tissues demonstrated more rapid and robust cell division, ultimately leading to whole-plant regeneration [19]. This study represents the inaugural attempt to isolate protoplast from callus cultures of *C. sativa*, outlines the improvements in yield offered by AIP application in this species, and reports the initial observation of cannabis protoplast division. 

Despite the success of callus induction methods across various genotypes [6], inducing cell differentiation and regeneration in cannabis callus cultures remains challenging. Protoplast yields in cannabis have begun to catch up with the model species Arabidopsis and members of the Solanaceae family [11,12], necessitating the refinement of existing isolation methods for specific research areas such as whole-plant regeneration [7,11,14]. Given recent reports that cannabis hypocotyls have higher regenerative potential than other tissues [13,14,27], this study explores the juvenile hypocotyl-derived callus as a protoplast source for developing a protoplast-to-plant regeneration system. This study investigates the beneficial effects of AIP in callus culture medium in enhancing protoplast yield compared with AIP-free medium. The hypothesis that AIP inclusion would reduce total phenolics and tissue browning, which are associated with improved protoplast yield and division, was confirmed. Callus-derived protoplast yield increased by 334% as a result of the inclusion of AIP in the medium (LT-AIP, containing 1 mM AIP; Figure 1 and Figure 2). AIP’s use in making callus cultures amenable to protoplast isolation has been well studied in *Ulmus americana*, demonstrating increased protoplast yields and subsequent regeneration from leaf tissues through AIP’s competitive inhibition of the phenylpropanoid pathway [19,20]. 

This study utilized an enhanced enzyme composition reported by Beard et al. [7], incorporating pectolyase and significantly increasing protoplast yield compared with a pectolyase-free mixture. Although our achieved yield (8.78 × 10^4^) did not attain the levels reported by Beard et al. [7] (2.27 × 10^6^) or the more recent publication by Zhu et al. [12] (1.15 × 10^7^), the disparity may be attributed to using callus instead of leaf tissue and potential differences in genetic backgrounds, which have been reported to affect the in vitro responses of cannabis [5]. Despite lower yields, callus-derived protoplasts may be more suitable for specific applications, especially those related to protoplast-to-plant regeneration systems. The observed enhancements with AIP inclusion suggest that targeting cell wall composition provides another viable avenue for improving cannabis protoplast isolation methods, alongside recent protocol refinements [11,12]. 

In the present study, the accumulation of phenolic compounds was estimated using two established assays: the F-C assay and a “browning assay”, which gauges absorbance at 340 nm in aqueous extracts. These assays were chosen for their demonstrated reliability in assessing phenolic compound accumulation in plant tissues and their capacity to monitor relative changes resulting from treatment effects [21,28,29]. The assays consistently indicated that callus grown in AIP-containing medium accumulated fewer total soluble phenolics and exhibited reduced browning compared with the control group. Specifically, the F-C assay revealed that AIP-treated callus exhibited an estimated 28% reduction in soluble phenolic content compared with the control, while the browning assay similarly demonstrated a 52% decrease in browning in AIP-grown tissues compared with the control. A previous study in callus cultures of *Artemisia annua* grown on AIP also recorded a drop in phenolics as measured using these assays; however, the reductions in phenolics and browning were larger than the reductions reported here [21]. The larger reduction in phenolics and tissue browning in *Artemisia* callus compared with the present study is likely a result of numerous factors, such as the different starting material for the callus, different species, and different plant growth regulators (PGRs) used to induce and maintain callogenesis. Nevertheless, this discrepancy might also reflect the limitation of the assays employed for quantifying tissue browning and soluble phenolics. 

The F-C assay, an established method for quantifying phenolic content, has received validation by Association of Official Analytical Chemists (AOAC) International [30]. However, its indirect nature, measuring phenol oxidation under alkaline conditions and a subsequent reaction with the F-C reagent, can lead to underreported results due to variations in phenolic compound properties and potential interference from non-phenolic compounds, including aromatic amines, ascorbic acid, and proteins [31,32]. As a result, the F-C assay, which benefits from ease of use and affordability, may underestimate total phenolic compounds compared with a more targeted and costly approach guided by high-performance liquid chromatography (HPLC) [32]. In contrast, the browning assay, which directly measures phenolics based on their absorbance profiles, may be less susceptible to such interferences [21]. This limitation of the F-C assay could explain the observed 28% reduction in total soluble phenolics between LT-C and LT-AIP media compared with the 52% reduction in browning observed in the browning assay. Notably, the cultivation of callus in darkness reduces the likelihood of significant accumulation of interfering compounds, such as carotenoids and chlorophyll. These differences in assay sensitivity likely contribute to the weak-to-moderate correlation between yield and reduced phenolic content (Appendix A). Regardless, the two assays demonstrated similar trends and supported the overall hypothesis.

The impact of subculture frequency on protoplast yield and quantifiable phenolics, as assessed by both assays, was negligible, despite qualitative observations favoring weekly subcultures for callus health. This suggests that AIP bioavailability in the culture medium remained sufficient for continuous competitive inhibition over a two-week period. Alternatively, rapid AIP uptake, combined with prolonged in vivo persistence, likely resulted in elevated endogenous AIP levels. Past studies indicate that AIP-induced phenotypes revert when AIP exposure ceases, implying either AIP degradation or increased PAL production without AIP [19]. Radiolabelled AIP studies could provide insights into its in vivo behavior. The study demonstrated that 1 mM AIP exposure effectively reduced PAL activity in cannabis calli throughout the extended two-week culture.

Successful protoplast culture and regeneration conditions are highly species-specific, being influenced by factors including nutrient availability, PGRs, agar type/concentration, osmolarity, and cell density [7,10,14,19,24,33]. This study focused on the inclusion of AIP in culture media and protoplast density in low-melting-point agarose beads. Several studies have highlighted the importance of cell density for inducing and sustaining cell division and eventual regeneration in protoplast [19,24,34]. Here, we report the initial stages of protoplast division in cultures at a density of 2.0 × 10^5^ protoplasts/mL, but cell division was not observed at lower densities. Initial protoplast activity prior to division was characterized by irregular cell morphology such as bulging as well as the formation of pear- and peanut-shaped cells (Figure 3). Protoplasts cultured in KM5/5 medium with 10 µM AIP (cell density: 2.0 × 10^5^) exhibited increased initial activity. Previous work using the same culture system found this density optimal for cell division and regeneration in American elm [19]. 

In the current study, microcalli, or small clusters of dividing cells, were generated from protoplasts at a concentration of 2.0 × 10^5^ during the initial three weeks of culture. However, sustained cell division and viability were not observed beyond this period, and the reason for this decline remains unclear. Protoplast densities in the range of 10^5^ cells per mL are frequently cited as a favourable density for protoplast division [19,33,34,35]. The present study found that a density of 2.0 × 10^5^ resulted in initial cell divisions, but cell division was not sustained beyond three weeks, suggesting that cannabis may require altered conditions beyond three weeks to sustain microcalli development. The type of culture medium and the composition of PGRs also play pivotal roles in protoplast regeneration. In this study, only KM5/5 medium was explored for protoplast regeneration, and despite KM basal salts being known to promote protoplast growth at low densities [24], these results suggest that the formulation does not sustain cell division beyond the initial stages in cannabis or that other factors such as osmotic potential need to be altered as the calli develop. In addition, it is important to note that many basal salts commonly used in other species are not as successful in cannabis [1]. The presence of cellular debris in the final isolate might have adversely affected regeneration, emphasizing the need for improvements in gradient purification methods. Recent reports have described methods for enhanced digestion using vacuum infiltration and a modified gradient purification [11,12]. However, it is important to note that neither of these methods addresses the competence of cells for regeneration; instead, they focus on the use of protoplasts for transient gene expression. To date, this study represents one of the first demonstrations of protoplast cell division from cannabis callus-derived protoplasts. 

This study demonstrates the feasibility of isolating protoplasts from *C. sativa* callus cultures, coupled with a 334% enhancement in protoplast yield achieved through the incorporation of AIP in callus culture medium. AIP makes cell walls more susceptible to enzymatic degradation and protoplast isolation by reducing phenolic content and tissue browning, as confirmed by the F-C and 340 nm tissue browning assays. This suggests that AIP’s mode of action is conserved in cannabis. Notably, the two-phase culture system demonstrated cell division at a concentration of 2.0 × 10^5^ protoplasts per mL, signifying the first known report of cell division from cannabis protoplasts. These findings lay the groundwork for future development in the areas of cannabis breeding and biotechnology, such as the generation of interspecific hybrids, altered ploidy levels, and the incorporation of transgenes. 

## 4. Materials and Methods

### 4.1. Callus Culture Initiation

Callus cultures were established from etiolated hypocotyls of 3-week-old seedlings of *Cannabis sativa* cv. “Finola”. The seeds were surface-sterilized with a solution of 10% commercial bleach (5.25% sodium hypochlorite, Clorox, Brampton, ON, Canada) and 0.1% Tween 20 for 12 min, followed by three rinses with sterilized deionized water for 5 min, each in a laminar airflow cabinet (Design Filtration Microzone, Stittsville, ON, Canada). The surface-sterilized seeds were germinated on a growth medium based on the method described in Hesami et al., 2021 [36]. This medium included 0.43× MS basal salt mixture with the van der Salm Modification (M5541, PhytoTech Labs, Lenexa, KS, USA), 2.3% sucrose (*w*/*v*), and 0.6% agar (*w*/*v*) (Fisher Scientific, Hampton, NH, USA). The seeds were germinated in the dark for 3 weeks.

Etiolated hypocotyl segments from the germinated seedlings were used to initiate callus cultures. Callus induction medium (LT-C) consisted of MS (M524; Phytotechnology Laboratories) nutrients, 3% sucrose, 0.8% 132 type E agar (*w*/*v*) (Sigma Aldrich, St. Louis, MO, USA), 0.5 µM NAA (Sigma Aldrich), and 1.0 µM TDZ (Caisson Laboratories, Inc., Smithfield, UT, USA) adjusted to a pH of 5.7 [6,37] and a modified version referred to as LT-AIP that contained 1 mM AIP (hydrochloride salt form; AmBeed, Arlington Heights, IL, USA). Subsequently, 1 cm long hypocotyl segments were placed onto approximately 30 mL of either LT-C or LT-AIP medium in deep Petri dishes (VWR International, Mississauga, ON, Canada). These Petri dishes were sealed with PVC film and incubated in the dark at 25 °C in a controlled environment growth chamber for 6 weeks to induce callus formation. After 6 weeks, the callus cultures were subcultured and maintained for an additional 2 weeks, resulting in a total callus initiation period of 2 months. Following this 2-month period, the LT-C and LT-AIP calli were divided into two subgroups, each consisting of 5 replicates (Petri dishes; n = 5). Each replicate contained 4 pseudoreplicate callus cultures. These cultures were subcultured onto fresh media either weekly (every 7 days) or biweekly (every 14 days) for 2 weeks before being used for protoplast isolation.

### 4.2. Enzyme Preparation for Protoplast Isolation

The enzyme solution used for protoplast digestion was freshly prepared for each digestion, following the procedure described by Yoo et al. [25] with slight modifications. Briefly, a solution containing 20 mM MES, 0.4 M mannitol, and 20 mM KCl was prepared using stock solutions [25] and adjusted to a pH of 5.7 with 1 M KOH/HCl. The solution was then transferred to a beaker and gently heated to 70 °C with stirring for 3–5 min. Afterward, the temperature was reduced to 55 °C, and cell-wall-digesting enzymes were added gradually until fully dissolved. The enzyme solution comprised 1.25% (*w*/*v*) Cellulase “Onozuka” R-10 (Yakult Pharmaceutical Ind. Co., Ltd., Tokyo, Japan), 0.3% (*w*/*v*) Macerozyme R-10 (Yakult Pharmaceutical Ind. Co., Ltd.), and 0.075% (*w*/*v*) Pectolyase Y-23 (Kyowa Chemical Products Co., Ltd., Osaka, Japan), following the method established by Beard et al. [7]. The solution was incubated at 55 °C for 10 min to inactivate DNases and proteases and improve enzyme solubility. Subsequently, it was cooled to room temperature, and an aqueous solution of calcium chloride dihydrate (1 M) was added to reach a final concentration of 10 mM. Finally, bovine serum albumin (BSA) was added from a 10% (*w*/*v*) stock to achieve a final concentration of 0.1%. The enzyme solution was then adjusted to a final volume of 60 mL using distilled water and filter-sterilized with a 0.2 µm nylon syringe filter.

### 4.3. Protoplast Digestion and Assessment

To evaluate the impact of AIP and subculture frequency on protoplast yield, a series of digestions were conducted, and the resulting protoplast yields were quantified. For each digestion, approximately 100 mg of the freshest, watery, pale-to-cream-colored callus was digested in 5 mL of the enzyme solution. An additional 100–300 mg of the same callus was set aside for total phenol analysis (as outlined below). The selected tissue was transformed into a soft paste by finely chopping the callus with sterile razor blades. The tissue slurry was then subjected to enzymatic digestion, which took place in 6-well, cell-culture grade, sterile polystyrene plates kept in the dark for 16 h on an orbital shaker. The shaker operated at 75 rpm at an ambient temperature of 25 °C. Digestions were performed for all 5 replicates of each treatment and were repeated on a second day two weeks later, resulting in a final sample size of n = 10. Each replicate represented a digestion carried out with callus sampled from a unique callus culture. After the 16 h digestion period, the digests were examined under an inverted microscope (Axiovert 200; Carl Zeiss Canada Ltd., Toronto, ON, Canada) to check for contamination and overall quality. Subsamples from each digestion were evaluated for protoplast density using a hemocytometer with a compound light microscope.

### 4.4. Protoplast Isolation and Culture

Callus tissues from each treatment were prepared and digested as described above. Each digestion was conducted in a 10 mm Petri dish (VWR International) using 15 mL of enzyme solution per 300 mg of callus tissue. After a 16 h digestion, protoplast purification was performed following the method outlined by Beard et al. [7] with modifications outlined in detail in the Appendix A. Briefly, Petri dishes were examined to confirm digestion and check for contamination. The enzyme solution was then filtered twice to remove large debris, before undergoing centrifugation. The pellet was resuspended in a matrix solution and protoplast purification employed a 60% iodixanol density gradient (outlined in Appendix A and shown in Appendix A). The protoplast band was collected, and a 30 µL aliquot was sampled for counting using a hemocytometer. Protoplast yield was determined as protoplasts per gram of fresh weight. Viability assessment involved staining with fluorescein diacetate (FDA) and counting live protoplasts under an epifluorescence microscope. Cells were stained by adding 60 µL/mL of 2 mg/mL FDA (Sigma-Aldrich) and incubating samples in the dark for approximately 10 min (Figure 1C). The protoplast band was suspended in KM5/5 medium [24], pH 5.7, and centrifuged. After removing the supernatant, the protoplast pellet was resuspended at twice the target culture density using KM5/5 medium. This suspension was mixed with an equal volume of 1.6% low-melting-point SeaPlaque agarose solution and transferred dropwise to a 6-well tissue culture-grade polystyrene plate. After solidification (~20 min), sterile KM5/5 medium (with or without 10 µM AIP) was added. Cultures were maintained in the dark at 25 °C and monitored for cell division, with various protoplast concentrations in the low-melting-point agarose beads tested, ranging from 0.5 × 10^5^ to 2.0 × 10^5^ protoplasts/mL. Cultures were stained for cell walls with 0.1 mg/mL calcofluor white dissolved in acetone (Sigma-Aldrich) for 10 min at the end of the culture period (3 weeks; Figure 3). FDA and calcofluor-stained cells were observed using an inverted epi-fluorescence microscope using a DAPI/Hoescht/AMCA filter set (Chroma, Bellows Falls, VT, USA).

### 4.5. Total Phenols and Browning Assay Extract Preparation

Extract preparation was adapted from the methods presented in Jones and Saxena [21]. Callus samples from weekly and biweekly subcultured cultures on LT-C and LT-AIP media were selected using the same criteria as for digestions. Samples ranged from 100 to 300 mg and were weighed with an analytical balance (Quintix^®^ 124-1S, Sartorius, Göttingen, Germany) and then flash-frozen with liquid nitrogen. For each treatment, callus sampling occurred on two separate occasions, totalling 10 replicates. Flash-frozen tissues were lyophilized for 24 h (FreeZone 4.5 L Model 77510; Labconco, Kansas City, MO, USA). After lyophilization, the dry weight was recorded with the same analytical balance. Samples were finely ground in 2 mL snap-cap microcentrifuge tubes using a SPEX SamplePrep 1600MiniG^®^ bead homogenizer (SPEX, Metuchen, NJ, USA) at 1000 rpm for 40 s. An aliquot of extraction solvent (1:1:1 distilled water/methanol/acetone by volume) was added to each tube, maintaining a 1:10 dry weight-to-solvent ratio. Samples were vortexed for approximately 30 s and sonicated for 1 h in an ice-cooled water bath at 27 ± 3 °C. Tubes were then centrifuged for 6 min at 17,700× *g* (Accupsin Micro 17, Fisher Scientific). The supernatant was transferred to a new 0.5 mL snap-cap microcentrifuge tube, re-centrifuged to remove any particulates, and collected for phenolic assays. 

### 4.6. Total Phenols and Browning Assay

Total phenols in callus was assessed using a modified Folin–Ciocalteu (F-C) assay with a gallic acid (Sigma-Aldrich) standard curve [21,30,38]. The analysis closely followed the method of Jones and Saxena [21], with minor adjustments outlined in detail in the Appendix A. In short, a gallic acid stock solution (2 mg/mL) was prepared, and a 7-point standard curve (50 to 1000 µg/mL) was established (Appendix A). In a 96-well microplate, F-C reagent and the extract were added, followed by aqueous 0.25 M Na_2_CO_3_. The microplate was incubated and mixed, and absorbance was measured at 765 nm [30]. 

Tissue browning was estimated by measuring absorbance at 340 nm as previously reported in Jones and Saxena [21]. Ferulic acid (Sigma-Aldrich) served as a standard, and a 7-point standard curve (50 to 1000 µg/mL) was established (Appendix A). In a 96-well plate (Corning Inc., Corning, NY, USA), extraction buffer, sample extracts, standards, or sample blanks were added. Absorbances at 340 nm were measured using a microplate spectrophotometer. Standard curves demonstrated strong linearity between 50 µg/mL and 1000 µg/mL in both assays. For detailed procedures, refer to the Appendix A.

### 4.7. Experimental Design and Statistical Analysis

For digestion efficiency assessment, a two-way cross-classified factorial design with complete randomization was utilized. Fixed effects of subculture frequency and the presence of AIP in the media on protoplast yield were investigated, along with their interaction. The statistical model used was
y=μ+frequency of subculture+AIP+frequency of subculture×AIP+e
where

y: measured response variables (protoplast yield, total soluble phenolics, and browning);µ: overall mean of the response variable;frequency of subculture and AIP: fixed effects;e: residual error.

Protoplast yield determinations were performed twice, two weeks apart, resulting in 10 replicates per treatment (n = 10). SAS Studio software (v9.4, SAS Institute Inc., Cary, NC, USA) was used for statistical analyses. PROC GLIMMIX with a negative binomial distribution and a log link function was employed for ANOVA, and LSMEANS statement (α = 0.05) was used for mean comparisons. Missing data were represented as “.” in the dataset and were excluded from PROC GLIMMIX. Tukey–Kramer tests were employed to address multiple comparisons, and Microsoft Excel (Microsoft Corp., Redmond, WA, USA) was used for data visualization.

Optical density readings for total soluble phenolics and browning were quantified from regression curves generated from 7-point standard curves. ANOVA, the post hoc Tukey–Kramer test, and data normalization checks were performed for media and subculture frequency effects on total soluble phenolics and tissue browning. An R-side correction addressed non-homogenous variance in the browning assay. Kendall tau-b rank correlation between protoplast yield and total phenolics was obtained using SAS Studio software (v9.4). For standard curves, see the Appendix A.

## Figures and Tables

**Figure 1 plants-13-00130-f001:**
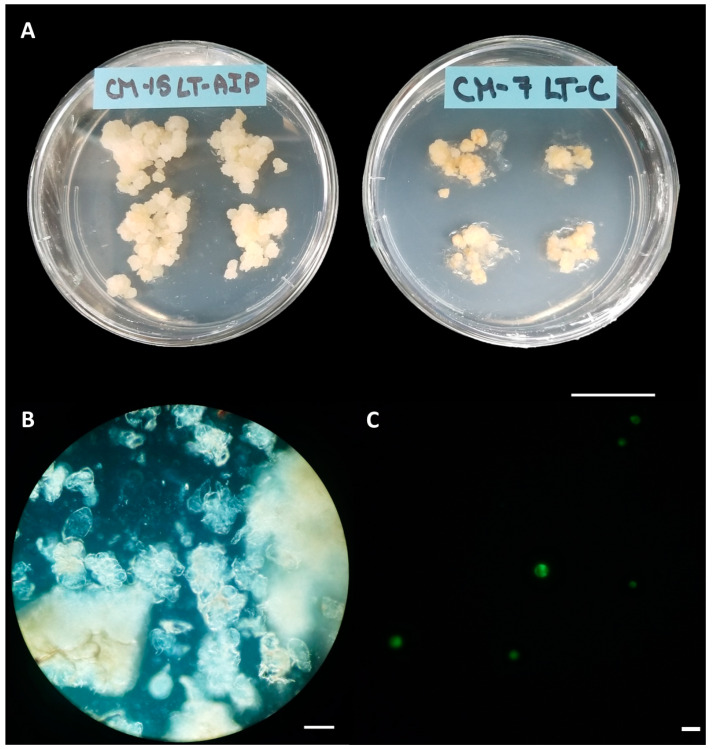
Callus cultures, digests, and viability. (**A**) Callus cultures grown on LT-AIP medium (left; callus induction medium + 1 mM AIP) showing larger and lighter tissues than those growing on LT-C medium (right; callus induction medium). Scale bar = 2.5 cm. (**B**) Digest of callus from LT-C, subcultured weekly; no protoplasts are visible. Scale bar = 100 µM. (**C**) Digest of callus from LT-AIP, subcultured weekly, stained with fluorescein diacetate (FDA). Protoplasts fluorescing green indicate viability. Scale bar = 50 µM.

**Figure 2 plants-13-00130-f002:**
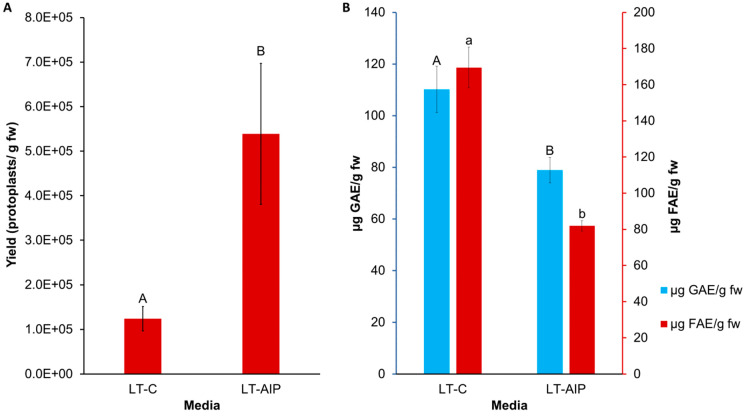
Comparison of LT-C and LT-AIP media with respect to protoplast yield and phenolic accumulation in callus tissues. (**A**) Protoplast yield from callus cultures grown on LT-C and LT-AIP media pooled from both 1- and 2-week subculture frequencies (n = 20). ANOVA revealed significant media effects (*p* = 0.0305). Yield reported as protoplasts/gram fresh callus weight (fw). Bars sharing the same letter were not significantly different at *p* ≤ 0.05 as determined by a Tukey-Kramer multiple comparisons test. Error bars represent standard error of the mean. (**B**) F-C assay (blue) and browning assay (red) averages compared between media types. Total soluble phenolics (µg of GAE—gallic acid equivalents) and tissue browning (µg of FAE—ferulic acid equivalents) at an optical density of 340 nm (OD340) in callus cultures on AIP-free (LT-C) and AIP-containing media (LT-AIP) pooled from both subculture frequency groups (n = 18). Error bars indicate standard error of the mean phenolic concentration for each media treatment, quantified in GAE or FAE. Bars sharing the same letter and case were not significantly different at *p* ≤ 0.05 as determined by a Tukey-Kramer multiple comparisons test.

**Figure 3 plants-13-00130-f003:**
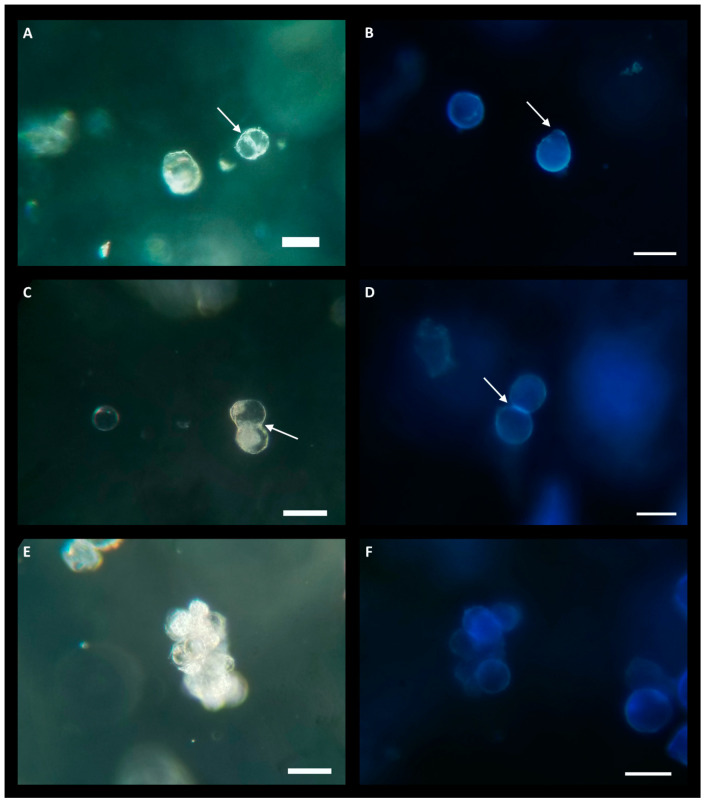
Protoplasts at various growth stages while cultured in low-melting-point agarose beads consisting of KM5/5 +10 µM AIP media at a density of 2.0 × 10^5^ protoplasts/mL. (**A**,**C**,**E**) are imaged under bright field. (**A**,**C**) were taken 6 days post-culture and (**D**) was taken 19 days post-culture. (**B**,**D**,**F**) are stained with calcofluor white, 3 weeks post-culture; blue florescence indicates cell wall formation. (**A**) The white arrow indicates the nascent cell wall during protoplast division. (**B**) Two protoplasts with blue florescence indicating the regeneration of cell walls. The white arrow indicates the location of the protoplast bulging, indicative of the beginning of cell division. (**C**) Protoplast undergoing division as seen by the formation of a cell wall (white arrow) next to an undivided protoplast (center-left). (**D**) Calcofluor stain of a dividing protoplast; the white arrow indicates bright-blue fluorescence at the formation of a new cell wall in the daughter cells. (**E**) Center shows a microcallus produced from successive protoplast divisions. (**F**) Calcofluor-stained microcallus. Scale bar = 50 µm.

**Table 1 plants-13-00130-t001:** Yield and viability pooled from weekly and biweekly digests used for determining viability and protoplast isolation for culture. Digests were made using 600 mg of tissue per 30 mL of enzyme digest. Viability was determined from FDA stain of cells.

Media	Yield (protoplasts/g fw)	Viability (%)
LT-C	2.87 × 10^4^ ± 1.10 × 10^4^	95.5 ± 2.6
LT-AIP	8.78 × 10^4^ ± 3.9 × 10^4^	92.1 ± 5.3

## Data Availability

All scripts used for data analysis have been uploaded to the Open Science Framework (OSF; OSF.io) under the DOI 10.17605/OSF.IO/N9YJU at the following link: https://osf.io/n9yju/?view_only=16ed7dc2432a465499f5d31cc0c1a12c.

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
