# Peer review of "Enhancing Protoplast Isolation and Early Cell Division from Cannabis sativa Callus Cultures via Phenylpropanoid Inhibition"

_plants, 2024, doi:10.3390/plants13010130_

Round 1
Reviewer 1 Report
Comments and Suggestions for Authors
In this manuscript (plants-2666046) entitled "2-aminoindane-2-phosphonic acid (AIP) improves Cannabis sativa protoplast isolation and early cellular division" submitted to Plants, Adrian S Monthony and Andrew Maxwell Phineas Jones have analyzed the efficacy of using a callus culture system, as an abundant source for protoplast isolation and lays the groundwork for a protoplast-to-plant regeneration system. Using hypocotyl-derived callus cultures, the efficacy of protoplast isolation and initial cell division were assessed. This research is interesting and convincing, but minor points need to be addressed to improve the quality of this manuscript.
1. Authors employed AIP at 1 mM in this study. What is the situation for other concentrations of AIP. At least three different concentrations of AIP should be examined in the revision.
2. For Figure 1c, protoplast viability should be examined by staining and microscopic observation in the revision. From the present Figure 1c, it is difficult to judge the protoplast.
3. For Figure 3, cell dividion should be examined by staining and microscopic observation in the revision. From the present Figure 3c and 3d, it is difficult to judge the cell division.
4. Full names of the abbreviations CRISPR, CBP, LT-C, LT-AIP, GAE, FAE, F-C, PGR, and AOAC should be spelt out at their first appearance in the revised manuscript.
Reviewer 2 Report
Comments and Suggestions for Authors
In the manuscript, the authors have presented the role of 2-aminoindane-2-phosphonic acid (AIP) in improving Cannabis sativa protoplast yield and its impact on early cellular division. The manuscript is well written, with minor typos and grammatical errors that must be fixed.
Line # 384-refer to Section Error! Reference source not found.).
The authors have highlighted the importance and major bottlenecks faced while using the protoplast system. The results presented are encouraging, and more studies in the future could address the factors preventing microcalli division and recovery of shoots, yielding a promising gene delivery system in Cannabis.
Comments on the Quality of English LanguageTypos and a few grammatical errors need to be fixed
Reviewer 3 Report
Comments and Suggestions for Authors
This paper on protoplast isolation from Cannabis sativa hypocotyls was interesting.
The paper could be greatly improved by substantial editing. There is too much repetition in the introduction and particularly in the Discussion. The reader gets that it is challenging!! The whole paper could be more succinct.
For the methods much of the detail would be better in the Supplementary section. For example, it is not necessary to have details on how to make a standard curve.
For Figure 1 C the circled protoplast are not really visible and I would question the value of this figure. For Figure 3, A B and C the 6-day post culture conditions are repeated 3 times and the only difference for Figure 3D is that it is 19-days post culture. The media and density do not need to be repeated for each photo.
There are minor grammatical errors but as the whole manuscript needs to be edited, I am sure these could be picked up without me noting them by line number.
